# Lipid Profile, Lp(a) Levels, and HDL Quality in Adolescents with Down Syndrome

**DOI:** 10.3390/jcm11154356

**Published:** 2022-07-27

**Authors:** Aleksandra Krzesińska, Anna Kłosowska, Kornelia Sałaga-Zaleska, Agnieszka Ćwiklińska, Agnieszka Mickiewicz, Gabriela Chyła, Jolanta Wierzba, Maciej Jankowski, Agnieszka Kuchta

**Affiliations:** 1Department of Clinical Chemistry, Medical University of Gdańsk, 80-211 Gdańsk, Poland; aleksandra.krzesinska@gumed.edu.pl (A.K.); kornelia.salaga-zaleska@gumed.edu.pl (K.S.-Z.); agnieszka.cwiklinska@gumed.edu.pl (A.Ć.); gabriela.chyla@gumed.edu.pl (G.C.); maciej.jankowski@gumed.edu.pl (M.J.); 2Department of Paediatrics, Haemathology and Oncology, Medical University of Gdańsk, 80-211 Gdańsk, Poland; anna.klosowska@gumed.edu.pl; 31st Department of Cardiology, Medical University of Gdańsk, 80-211 Gdańsk, Poland; agnieszka.mickiewicz@gumed.edu.pl; 4Department of Internal and Pediatric Nursing, Institute of Nursing and Midwifery, Medical University of Gdańsk, 80-211 Gdańsk, Poland; jolanta.wierzba@gumed.edu.pl

**Keywords:** lipid profile, lipoprotein(a), high-density lipoprotein, Down syndrome, cardiovascular disease, cardiovascular risk

## Abstract

The improvement in the lifespan of individuals with Down syndrome (DS) has created interest in the context of the development of age-related diseases. Among them is atherosclerosis-based cardiovascular disease (CVD), which seems to be an especially urgent and important issue. The aim of the present study was to evaluate the lipid markers that may clarify cardiovascular risk profiles in individuals with DS. To this end, we analyzed lipid profile parameters, including lipoprotein(a) (Lp(a)) levels, protein composition, and the antioxidative properties of high-density lipoprotein (HDL), in 47 adolescents with DS and 47 individuals without DS. Compared with the control group (C), subjects with DS had significantly increased concentrations of low-density lipoprotein cholesterol (105 ± 31 vs. 90 ± 24 mg/dL, *p* = 0.014), non-high-density lipoprotein cholesterol (120 ± 32 vs. 103 ± 26 mg/dL, *p* = 0.006), and triglycerides (72 [55–97] vs. 60 [50–77] mg/dL, *p* = 0.048). We found that patients with DS were characterized by significantly higher Lp(a) levels (31.9 [21.5–54.3] vs. 5.2 (2.4–16.1) mg/dL, *p* < 0.001). In fact, 57% of individuals with DS had Lp(a) levels above 30 mg/dL, which was approximately four times higher than those in the control group (DS 57% vs. C 15%). Apart from decreased high-density lipoprotein cholesterol levels in the subjects with DS (53 ± 11 vs. 63 ± 12 mg/dL, *p* < 0.001), differences in parameters showing the quality of HDL particles were observed. The concentrations of the main proteins characterizing the HDL fraction, apolipoprotein A-I and apolipoprotein A-II, were significantly lower in the DS group (144 ± 21 vs. 181 ± 33 mg/dL, *p* < 0.001; 33 ± 6 vs. 39 ± 6 mg/dL, *p* < 0.001, respectively). No significant differences between the groups were observed for the concentration of paraoxonase-1 (DS 779 ± 171 vs. C 657 ± 340 ng/mL, *p* = 0.063), enzyme activities toward paraoxon (DS 219 [129–286] vs. C 168 [114–272] IU/L, *p* = 0.949), or phenyl acetate (DS 101 ± 20 vs. C 93 ± 21 kIU/L, *p* = 0.068). There were no differences in myeloperoxidase activity between the study groups (DS 327 [300–534] vs. C 426 [358–533] ng/mL, *p* = 0.272). Our results are the first to demonstrate an unfavorable lipid profile combined with higher Lp(a) levels and quality changes in HDL particles in individuals with DS. This sheds new light on cardiovascular risk and traditional healthcare planning for adolescents with DS.

## 1. Introduction

Down syndrome (DS), the most frequent chromosomal abnormality that causes intellectual disability, occurs in between 1 and 1000–1100 live births worldwide [1]. The extra chromosome 21, or at least a portion of it, is related to intellectual disability and to several clinical alterations that manifest through the accelerated aging of different organs and tissues [1,2]. Advancements in medical healthcare have led to an unusual improvement in the life expectancy of individuals with DS, with an estimated mean survival age approaching age 60 [3]. The increased lifespan of people with DS has also changed the incidence of chronic disease, which has created concern for their long-term health, particularly in the context of the occurrence of atherosclerosis-based cardiovascular disease (CVD) [3,4].

Patients with DS are considered protected from atherosclerotic disease, and some researchers have proposed DS as an “atheroma-free” model of disease based on postmortem studies that found individuals with DS to have no atherosclerotic plaques, a decreased frequency of arteriosclerosis, or a decreased total area of raised lesions compared with individuals without DS [3,5]. However, more recent epidemiological studies have revealed increased mortality from CVD in the population with DS and have noted the need to better define risk factors in this group of patients [3,6].

Few equivocal studies have investigated the basic cardiovascular lipid risk factors in individuals with DS, and their results are not clear. Most studies indicate a coexistence of DS with an unfavorable lipid profile, including an increased concentration of triglycerides (TG) and decreased high-density lipoprotein cholesterol (HDL-C) [1]. Some studies indicate levels considered normal for these parameters [7]. To the best of our knowledge, there are no studies concerning the advanced aspects of lipoprotein disturbances. Lipoprotein fractions are not homogeneous groups of particles, and their importance in the development of atherosclerosis depends not only on their measurement assessed by cholesterol levels but also on their lipid and protein composition [8].

Among the lipoprotein particles involved in the pathogenesis of CVD, lipoprotein(a) (Lp(a))—a genetic, independent, and likely causal risk factor for CVD—requires special attention [9,10]. Lp(a) consists of a large glycoprotein apolipoprotein(a) (apo(a)), which is covalently attached to the apolipoprotein B-100 moiety of low-density lipoprotein (LDL) by a single disulfide bond. Lp(a), by its preferential binding to macrophages, and proinflammatory and pro-oxidative capacities, may initiate foam cell formation and, as a result, the deposition of cholesterol in atherosclerotic plaques [11]. As serum Lp(a) levels are controlled by the *LPA* gene on chromosome 6, they are relatively stable over a lifetime [12]. It is recommended to take a once-in-a-lifetime Lp(a) measurement, with possible confirmatory repeat measurements in those with very high concentrations [13]. It has been claimed that CVD risk is increased by approximately twofold for individuals with Lp(a) levels above 30 mg/dL [14]. Recent data indicate that the association between Lp(a) concentration and CVD risk is linear [15].

High-density lipoprotein (HDL) particles are considered atheroprotective and have been associated with a reduced risk of CVD [16]. The qualitative composition of HDL particles determines their capacity to promote cholesterol efflux from peripheral tissues, and individual HDL particles have different antioxidative properties [17]. The functionality of HDL particles may be impaired, which is associated with a change in their protein composition, including apolipoprotein A-I (ApoA-I) and paraoxonase-1 (PON-1) [17]. Human PON-1 is an HDL-related ester hydrolase enzyme that protects LDL and cell membranes from oxidation through the hydrolysis of biologically active lipid peroxides. Therefore, the anti-atherogenic property of HDL may be explained by PON-1 [17,18,19]. As numerous studies show, reduced PON-1 activity is characteristic in patients with atherosclerosis and is associated with increased myeloperoxidase (MPO) activity, resulting in the formation of dysfunctional HDL particles [20].

To the best of our knowledge, there are no studies assessing HDL quality and Lp(a) levels in individuals with DS, which, due to the significant prolongation of the lifespan of individuals with DS and the ambiguity in atherosclerosis risk assessments, seem to be worth analyzing. The aim of this study was to compare the lipid parameters, protein composition, and antioxidative properties of HDL, and Lp(a) levels in adolescents with DS and healthy individuals. 

## 2. Materials and Methods

### 2.1. Patients 

The study group consisted of 47 adolescents with DS, aged 9–18, who were under the care of the Pediatric Clinic at the University Clinical Center of the Medical University of Gdansk (Poland). The control group consisted of 47 healthy adolescents, aged 9–18, who were recruited from the outpatient General Practitioner Clinic at the same institution. The general characteristics of the study subjects are shown in Table 1. The analyzed groups were matched in terms of age and gender. 

The inclusion criteria were as follows: trisomy of chromosome 21 for the DS group; and for both groups, subjects (a) were older than 9 years of age; (b) had the signed informed consent of their parents for participation in the study; (c) lacked severe concomitant diseases; (d) had no obesogenic drugs in the medical interview; and (e) were willing to cooperate. The exclusion criteria were as follows: mosaic DS or translocation DS for the DS group; and for both groups, (a) the presence of severe associated diseases affecting energy balance (including diabetes, celiac disease, and renal failure); (b) inflammation assessed on the basis of C-reactive protein (level >3.10 mg/L) and morphology; (c) decompensated thyroid disease (thyroid-stimulating hormone level >4.94 µU/mL); and (d) a history of cancer or intestinal anomalies requiring surgical medical intervention.

The study was performed in accordance with the ethical guidelines of the 1975 Declaration of Helsinki (as revised in 2013). All study procedures were reviewed and approved by the Independent Bioethics Committee for Scientific Research at the Medical University of Gdansk (No. NKBBN/105-96/2016). 

### 2.2. Laboratory Measurements

Peripheral fasting blood samples were drawn between 7 a.m. and 8 a.m. The serum was separated after centrifugation at 1000× *g* for 15 min and was stored at −80 °C pending analysis. 

TC and TG were measured in serum using standard enzymatic colorimetric tests (Pointe Scientific, Warsaw, Poland). HDL was isolated by the precipitation of apolipoprotein-B-containing lipoproteins with heparin and manganese chloride, and HDL-C was determined enzymatically in supernatant using a kit (Pointe Scientific, Warsaw, Poland). LDL-C concentration was calculated using the Friedewald formula, and nonHDL-C was calculated by subtracting HDL-C from TC. The apolipoprotein (ApoA-I, ApoA-II, and ApoB) serum concentrations were measured using the nephelometric method with antibodies obtained from Siemens Healthcare Diagnostics (Eschborn, Germany) on a Behring laser nephelometer. Lp(a) concentrations were determined using a commercially available immunoturbidimetric assay (Randox, Crumlin, UK). The PONase and AREase activities of PON-1 were analyzed in a serum based on paraoxon and phenyl acetate hydrolysis, respectively, according to the procedure described earlier [21]. The total PON-1 concentration and MPO were determined using enzyme immunoassay kits (Biorbyt, Cambridge, UK). Thiobarbituric-acid-reactive substance (TBARS) concentration was measured using fluorescence spectroscopy according to a previously described method [22].

### 2.3. Statistical Analysis

All statistical analyses were performed using STATISTICA software, version 10 (StatSoft, Warsaw, Poland). The Shapiro–Wilk test was used to assess the normality of the distribution of the variables. The continuous variables were expressed as mean ± standard deviation or as medians with 25th and 75th percentiles. To compare data between the two groups, Student’s *t*-test for Gaussian variables and the Mann–Whitney *U* test for non-Gaussian variables were used. Pearson’s chi-squared test was used to compare categorical variables. Univariate correlations were assessed using standardized Spearman coefficients. A *p*-value below 0.05 was considered to be statistically significant.

## 3. Results

As expected, the adolescents with DS were shorter and had a higher BMI compared with the control group (Table 1). The values of the lipid profiles of the control group and DS are reported in Table 2. The concentrations of total cholesterol (TC) were similar in both groups. Mean low-density lipoprotein cholesterol (LDL-C) and non-high-density lipoprotein cholesterol (nonHDL-C) concentrations were significantly higher in subjects with DS than in the control group. The TG concentrations were lower in the control group, although these differences were at the limit of statistical significance. Compared to the control group, adolescents with DS had a significantly decreased mean level of HDL-C. We did not find significant correlations between BMI and lipid profile parameters in the DS group. The difference in the serum concentration of ApoB between subjects with DS and the control group was not significant. The analysis of the ApoB/ApoA-I ratio showed a remarkable increase in subjects with trisomy of chromosome 21 (Table 2).

In our study, adolescents with DS, compared to the control group, were characterized by significantly higher Lp(a) levels (Figure 1). More than 50% had Lp(a) levels above 30 mg/dL, which was approximately four times higher than in the control group (Table 2).

No significant differences were observed in any of the lipid profile parameters or BMI among individuals with Lp(a) levels below and above 30 mg/dL in the control group, as well as in the DS group, as shown in Table 3. No significant correlations between Lp(a) levels and other lipoprotein parameters were found. The correlation of Lp(a) levels and BMI was not significant for both groups (C:R = 0.17, *p* = 0.265; DS:R = 0.18, *p =* 0.216).

Apart from decreased HDL-C levels in the subjects with DS, differences in parameters showing the quality of HDL particles were also observed. The concentrations of the main proteins characterizing the HDL fraction (ApoA-I and ApoA-II) were significantly lower in the DS group. In addition, the ApoA-I/ApoA-II ratio was significantly higher in individuals without DS (Figure 2).

The study subjects were characterized by a significantly different concentration of thiobarbituric-acid-reactive substance (TBARS), which was higher in adolescents with DS (Table 4). The univariate correlation analysis indicated that TBARS positively correlated with concentrations of TC, LDL, nonHDL, and Lp(a) in the entire study population (R = 0.36, *p* < 0.001; R = 0.34, *p* < 0.001; R = 0.35, *p* < 0.001; R = 0.27, *p* = 0.008, respectively). No significant differences between the groups were observed for the activity and concentration of PON-1 (Table 4). The correlation of PON-1 activity with the concentration of HDL-C and ApoA-I in the analyzed groups was irrelevant. As presented in Table 4, the PON-1/ApoA-I ratio was significantly higher in subjects with DS. There were no differences in MPO activity between both study groups.

Figure 3 shows the distribution of the HDL-related antioxidant enzyme PON-1 activity toward two substrates, namely, paraoxon for paraoxonase (PONase) activity and phenyl acetate for arylesterase (AREase) activity, in both groups. This relationship enabled the extraction of three groups (PON-1 phenotypes) of patients with different PONase versus AREase values (less than 1.5, between 1.5 and 4.0, and higher than 4.0). The number of phenotypes in the participants with DS and control participants was similar; the smallest number of participants had a PONase/AREase ratio above 4 (Table 5).

## 4. Discussion

For many years, CVD in individuals with DS has been considered a marginal problem [7]. However, with the improvement in life expectancy, an assessment of the risk and development of CVD in this group has become an urgent and important issue.

In our study, we found that adolescents with DS had more atherogenic lipoprotein particle profiles than adolescents without DS of comparable age and gender. Our study is the first to evaluate Lp(a) levels in individuals with DS and to indicate that these patients are characterized by higher Lp(a) levels, with a median value approximately six times higher than that in the control group. Lp(a) levels above 30 mg/dL, indicating twice the CVD risk [13], occurred in more than 50% of adolescents with DS and occurred significantly more frequently than in their peers without DS. Lp(a) levels are controlled by the *LPA* gene and are highly heritable, and many studies have attempted to detect the genomic regions affecting Lp(a) concentrations [23]. Many researchers who have investigated the Lp(a) trait have found a strong linkage signal at the *LPA* gene on chromosome 6, but no significant linkage results for chromosome 21 have been observed [24]. In agreement with these findings, there are no known genetic mechanisms responsible for elevated Lp(a) levels in individuals with DS characterized by trisomy of chromosome 21. We demonstrated no significant correlations between Lp(a) levels and other lipoprotein parameters. We also found no significant differences in any of the lipid profile parameters, including TC, LDL-C, HDL-C, nonHDL-C, and TG, in individuals with Lp(a) levels below and above 30 mg/dL in both study groups. These results suggest that Lp(a) is independent from other lipid parameters, which is in accordance with a previous publication by Genest indicating that 19% of patients with premature coronary heart disease had elevated Lp(a) levels, with 13% of them having no dyslipidemia [25]. Our results clearly show that subjects with DS have higher Lp(a) levels, and they indicate the need for further research to understand the pathophysiological role of Lp(a) in DS, which may help to elucidate the contribution of this quantitative trait to the risk of CVD.

An unfavorable lipid profile frequently seen in subjects with DS involves elevated levels of TG and decreased HDL-C [6]. Our results confirm these findings in adolescents with DS. We also demonstrated significantly higher mean LDL-C and nonHDL-C in subjects with DS and an apparently similar concentration of TC between the groups. In previous studies, disturbances in lipid profile parameters in subjects with DS have differed from each other. Buonouomo et al. found that Italian individuals (from 2 to 19 years old) were characterized by high levels of TC, LDL-C, and TG and low HDL-C, except for girls over 15 years of age, for whom normal values of these parameters were presented [1]. In a study by Draheim et al., only TG was increased with cholesterol concentrations within normal values [26]. In a study by Adelekan et al., lipid profile parameters were within the recommended range; however, they were still higher than the results of their peers without DS in the control group [3]. However, Dorner et al. indicated that adults with DS had lower cholesterol values than control group subjects [27]. The heterogeneity of the findings describing lipid profiles in subjects with DS may result from the different activities of studied populations, as well as diet. Our earlier paper showed that children with DS are characterized by abnormal eating habits that can significantly affect the level of cholesterol [28]. As in the studies cited above, we have no knowledge about the nutritional status or the level of physical activity of the compared populations.

Parallel to the differences presented in our study in the concentrations of LDL-C and HDL-C between the groups, we showed an increased ApoB/ApoA-I value in the DS group. Some researchers have indicated that the ApoB/ApoA-I ratio is the strongest indicator of the risk of CVD and may be particularly useful in assessing its risk in metabolic syndrome, even when concentrations of LDL are generally not elevated [29].

HDL particles protect against atherosclerosis mainly through their ability to promote cholesterol efflux from lipid-laden macrophages in the artery wall. HDLs also have several protective properties depending on their concentration expressed as HDL-C and the quantitative and qualitative composition of the lipoproteins [30]. There is a lack of current results assessing the quality and protein composition of HDL particles in DS. In our research, in addition to finding significantly decreased mean levels of HDL-C in adolescents with DS, we evaluated the concentrations of the main proteins characterizing the HDL fraction (ApoA-I and ApoA-II), and we observed significantly lower concentrations of ApoA-I, ApoA-II, and the ApoA-I/ApoA-II ratio in adolescents with DS. Although the physiologic anti-atherosclerotic effects of ApoA-I are well established, the function of ApoA-II is not fully elucidated. Some studies suggest that increased ApoA-II impairs reverse cholesterol transport and the antioxidant function of HDL and, as a result, promotes atherosclerosis [31]. A decreased ApoA-I/ApoA-II ratio in subjects with DS indicates an enrichment of HDL molecules with ApoA-II, which may affect the quality of these particles.

Besides ApoA-I, the antioxidative and anti-inflammatory properties of HDL particles are strictly connected to PON-1 activity [19]. In our study, we characterized PON-1 by quantifying its concentration and enzymatic activities toward two substrates: paraoxon and phenyl acetate. The distribution of the HDL-related antioxidant enzymes’ PON-1 activity enables the extraction of three groups (PON-1 phenotypes) of patients with different PONase/AREase values. Previous studies have revealed that the phenotype selected on the basis of the PONase/AREase ratio is closely related to a common PON-1 polymorphism: Q (Glutamine) or R (Arginine) at codon 192 [21]. Some data indicate that this may be relevant in CVD risk assessment [32,33,34]. In our study, the number of phenotypes in the participants with DS and control participants was similar, which may support the hypothesis that polymorphisms influence PON-1 activity; however, this is clearly not responsible for the anti-atherogenic properties of PON-1.

Despite significantly lower concentrations of HDL-C and ApoA-I, adolescents with DS did not differ significantly from the control group in terms of the concentration and activity of PON-1. Consequently, the PON-1/ApoA-I ratio, indicating the amount of enzyme per HDL particle, was significantly higher in the DS group, which is a surprising result and clearly shows the need for further research on the properties of lipoproteins in this group of patients. Many studies have indicated that PON-1 may be particularly sensitive to the oxidation conditions established by MPO enzymatic activity, which plays a crucial role in creating dysfunctional HDLs [17,19]. Interestingly, despite there being no differences in MPO activity between the study groups, adolescents with DS were characterized by significantly higher concentrations of TBARS, the marker of lipid peroxidation [22].

Considering the importance of metabolism and the quality of HDL particles in CVD and Alzheimer’s disease, and that approximately 50% of adults with DS develop Alzheimer’s disease by their 60s [6], the nature of HDL quality and PON-1 activity in subjects with DS is interesting and needs further investigation.

## 5. Conclusions

Our data demonstrated unfavorable lipid profiles in conjunction with significantly higher Lp(a) levels and quality changes in HDL particles in adolescents with DS compared to the control group. This may be valuable in providing information about additional cardiovascular risk profiles and may shed new light on traditional healthcare planning for people with DS. Further research is required to explain the pathomechanisms of lipoprotein metabolism and the quality of HDL particles and to establish the risk stratification and prognosis of cardiovascular complications in individuals with DS.

### 5.1. Strengths of the Study

Our study is the first to evaluate Lp(a) levels in individuals with DS and makes an important contribution to the traditional cardiovascular risk profile parameters in these subjects. Carrying out research on Lp(a) concentrations in the population with DS may be valuable to increase the understanding of further morbidity and mortality from CVD in this group. Despite the many studies that show unfavorable basic lipoprotein fractions in subjects with DS, the findings obtained by the researchers are inconclusive, and there has been a lack of current results assessing the qualitative aspects of lipoproteins. A strength of this study is also in the wide-ranging evaluation of the protein composition and antioxidative properties of HDL by assessing HDL-related enzymes in adolescents with DS.

Considering the long-term surveillance of individuals with DS, the results of our young study group indicate the necessity of monitoring cardiovascular-risk-related parameters at an early age and provide useful insights into clinical practice.

### 5.2. Limitations of the Study

A limitation of this study is the relatively small sample size. This is partly due to the number of patients who are treated at our center, the number of parents and children who consented to participate in the tests and blood collection, and the exclusion criteria that impacted the final number of study subjects. Another limitation of this study is that it did not match the groups in terms of height and BMI. In the general population, the relationship of BMI to lipid profile is well known. However, the mechanism by which individuals with DS develop a particular atherogenic lipid profile still remains unclear, but it is suggested that it does not appear to be elucidated by obesity, as measured by BMI [3], which seems to be in line with our results showing no significant correlations between BMI and lipid profile parameters in the DS group. Our results are preliminary results that emphasize the need for further research to clarify the pathomechanisms of lipoprotein metabolism in individuals with trisomy of chromosome 21.

## Figures and Tables

**Figure 1 jcm-11-04356-f001:**
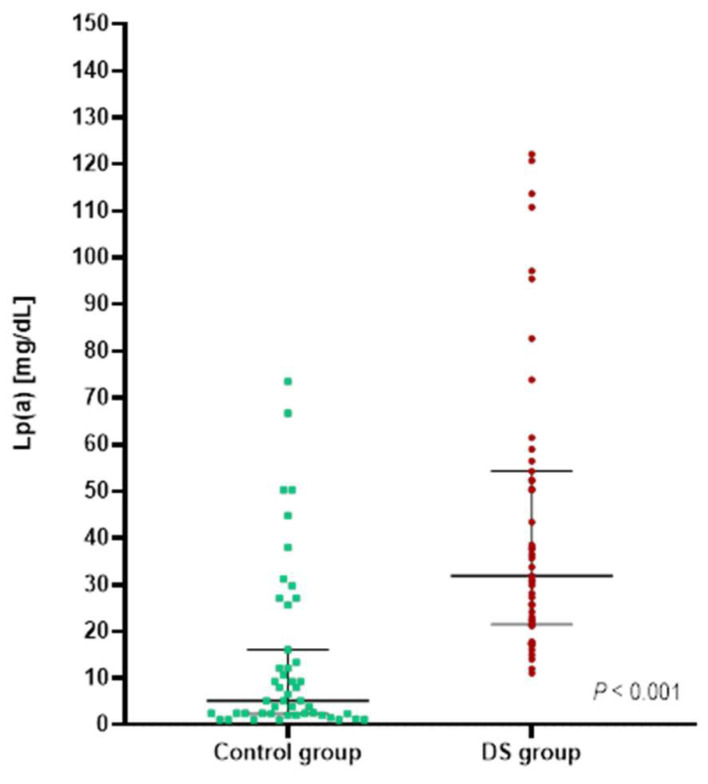
Lp(a) levels in patients with Down syndrome (DS group) and patients without Down syndrome (control group). Values are presented as medians (25–75th percentiles) and assessed using the Mann–Whitney *U* test.

**Figure 2 jcm-11-04356-f002:**
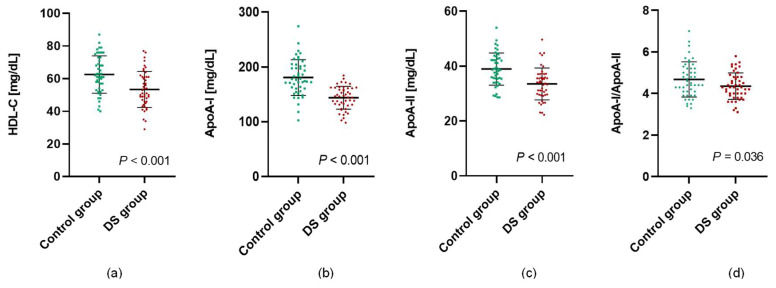
Concentrations of HDL (**a**), ApoA-I (**b**), ApoA-II (**c**), and ApoA-I/ApoA-II ratio (**d**) in patients with Down syndrome (DS group) and patients without Down syndrome (control group). Values are presented as mean ± standard deviation and were assessed using Student’s *t*-test.

**Figure 3 jcm-11-04356-f003:**
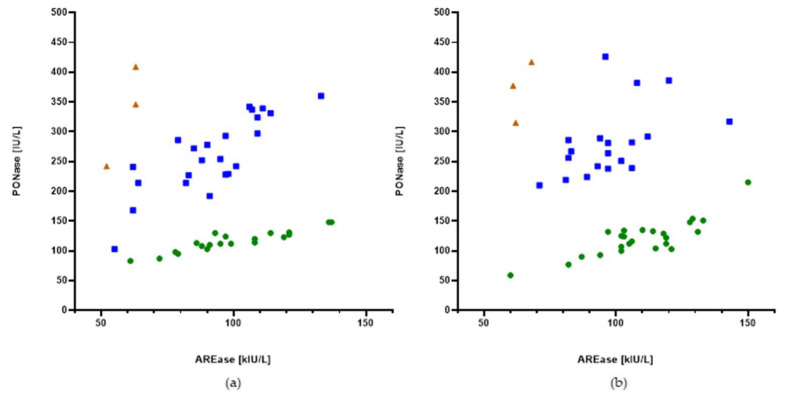
PONase activity toward AREase activity in the (**a**) control group and (**b**) DS group. Dots: PONase/AREase < 1.5 (green); squares: PONase/AREase = 1.5–4.0 (blue); and triangles: PONase/AREase > 4.0 (brown).

**Table 1 jcm-11-04356-t001:** Baseline characteristics of patients with (DS group) and without Down syndrome (control group).

	Control Group *N* = 47	DS Group *N* = 47	*p*-Value
**Gender, M/F (*N*)**	25/22	19/28	0.215 ***
**Age (years)**	14 (11–15)	14 (13–17)	0.072 **
**Weight (kg)**	50.6 ± 15.3	48.8 ± 12.6	0.541 *
**Height (m)**	1.58 ± 0.17	1.47 ± 0.11	<0.001 *
**BMI (kg/m^2^)**	19.8 (17.3–22.2)	21.4 (19.2–25.0)	0.008 **

Values are presented as mean ± standard deviation or as median (25th and 75th percentiles). * Student’s *t*-test; ** Mann–Whitney *U* test; *** Pearson’s chi-squared test. M—male; F—female; BMI—body mass index.

**Table 2 jcm-11-04356-t002:** Characteristics of the lipid parameters in the study groups.

	Control Group *N* = 47	DS Group *N* = 47	*p*-Value
**TC (mg/dL)**	166 ± 26	174 ± 35	0.236 *
**HDL-C (mg/dL)**	63 ± 12	53 ± 11	<0.001 *
**LDL-C (mg/dL)**	90 ± 24	105 ± 31	0.014 *
**TG (mg/dL)**	60 (50–77)	72 (55–97)	0.048 **
**nonHDL-C (mg/dL)**	103 ± 26	120 ± 32	0.006 *
**ApoA-I (mg/dL)**	181 ± 33	144 ± 21	<0.001 *
**ApoA-II (mg/dL)**	39 ± 6	33 ± 6	<0.001 *
**ApoB (mg/dL)**	65.5 ± 11.7	67.6 ± 12.3	0.391 *
**ApoA-I/ApoA-II**	4.68 ± 0.84	4.35 ± 0.64	0.036 *
**ApoB/ApoA-I**	0.37 ± 0.11	0.48 ± 0.09	<0.001 **
**Lp(a) (mg/dL)**	5.2 (2.4–16.1)	31.9 (21.5–54.3)	<0.001 **
**Lp(a) >30 mg/dL (*N*)**	7/15%	27/57%	<0.001 ***

Values are presented as mean ± standard deviation or as median (25th and 75th percentiles). * Student’s *t*-test; ** Mann–Whitney *U* test; *** Pearson’s chi-squared test. TC—total cholesterol; TG—triglycerides; HDL-C—high-density lipoprotein cholesterol; nonHDL-C—non-high-density lipoprotein cholesterol; LDL-C—low-density lipoprotein cholesterol; ApoA-I-apolipoprotein A-I; ApoA-II—apolipoprotein A-II; ApoB—apolipoprotein B; ApoA-I/ApoA-II—ApoA-I versus ApoA-II ratio; ApoB/ApoA-I—ApoB versus ApoA-I ratio; Lp(a)—lipoprotein(a).

**Table 3 jcm-11-04356-t003:** Lipid profile parameters and BMI in the context of Lp(a) levels in study population.

	Control Group		*p*-Value	DS Group		*p*-Value
	Lp(a) < 30 mg/dL *N* = 40	Lp(a) > 30 mg/dL *N* = 7		Lp(a) < 30 mg/dL *N* = 20	Lp(a) > 30 mg/dL *N* = 27	
**TC (mg/dL)**	163 ± 22	182 ± 40	0.067 *	177 ± 37	172 ± 33	0.611 *
**HDL-C (mg/dL)**	62 ± 12	63 ± 10	0.961 *	57 ± 11	51 ± 11	0.054 *
**LDL-C (mg/dL)**	87 ± 22	104 ± 31	0.093 *	104 ± 34	105 ± 29	0.862 *
**TG (mg/dL)**	59 (50–72)	64 (46–85)	0.788 **	72 (55–105)	73 (55–97)	0.923 **
**nonHDL-C (mg/dL)**	101 ± 23	119 ± 39	0.079 *	120 ± 35	121 ± 30	0.904 *
**BMI (kg/m^2^)**	19.7 (17.3–22.1)	20.0 (15.5–26.8)	0.427 **	21.0 (18.8–24.6)	21.7 (19.6–25.9)	0.349 **

Values are presented as mean ± standard deviation or as median (25th and 75th percentiles). * Student’s *t*-test; ** Mann–Whitney *U* test.

**Table 4 jcm-11-04356-t004:** Characteristics of the oxidative-stress-related parameters in patients with and without Down syndrome.

	Control Group	DS Group	*p*-Value
**PONase (IU/L)**	168 (114–272)	219 (129–286)	0.949 **
**AREase (kIU/L)**	93 ± 21	101 ± 20	0.068 *
**PON-1 (ng/mL)**	657 ± 340	779 ± 171	0.063 *
**PON-1/ApoA-I**	3.7 (1.5–5.0)	5.5 (4.5–6.5)	<0.001 **
**TBARS (µmol/L)**	4.5 (2.9–6.2)	7.3 (5.1–8.9)	<0.001 **
**MPO (ng/mL)**	426 (358–533)	327 (300–534)	0.272 **

Values are presented as mean ± standard deviation or as median (25th and 75th percentiles). * Student’s *t*-test; ** Mann–Whitney *U* test. PONase—paraoxonase activity; AREase—arylesterase activity; PON-1—paraoxonase-1 concentration; PON-1/ApoA-I—PON-1 versus ApoA-I ratio; TBARS—thiobarbituric-acid-reactive substance; MPO—myeloperoxidase.

**Table 5 jcm-11-04356-t005:** PON-1 phenotypes frequencies.

	Control Group			DS Group		
**PONase/AREase**	**<1.5**	1.5–4	>4	<1.5	1.5–4	>4
** *N* **	**20**	24	3	24	20	3
**PONase/AREase**	**1.2 (1.1–1.3)**	2.8 (2.5–3.1)	5.5 (4.7–6.5)	1.1 (1.0–1.2)	2.8 (2.6–3.2)	6.1 (5.3–6.2)
**PONase (IU/L)**	**114 (106–129)**	253 (221–311)	346 (242–409)	123 (104–134)	266 (239–291)	377 (315–417)
**AREase (kIU/L)**	**100 ± 21**	92 ± 19	59 ± 6	110 ± 19	99 ± 17	64 ± 4

Activities of paraoxonase and arylesterase, as well as PONase/AREase ratio, are presented as mean ± standard deviation or as median (25th and 75th percentiles).

## Data Availability

The data presented in this study are available on request from the corresponding author.

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
