# Peer review of "Lipid Profile, Lp(a) Levels, and HDL Quality in Adolescents with Down Syndrome"

_jcm, 2022, doi:10.3390/jcm11154356_

Round 1

Reviewer 1 Report

This is a clinical relevant study describing the abnormalities in serum lipid profiles in adolescent patients with Down syndrome (DS) as compared with non-DS healthy persons. The data is clean and straightforward and adds some new important data on the possible atherogenic lipid profile of DS. Only minor changes are suggested

-  A minor comment regarding the possible causes for the heterogeneity of previous findings describing both lower and higher levels of total cholesterol values as compared do healthy counterparts  - would add value (As a continuation of paragraph ending at Line 280)

-  The issue of BMI should be better discussed in the limitations as BMI does influence lipid profile and components of HDL chol.  An analysis with adjustments for BMI (or BMI z score) would have been useful.

-  TBARS – needs to be explained – line 149

Reviewer 2 Report

In this study KrzesiÅ„ska et al. set out to investigate the lipidemic profile of patients with Down Syndrome (DS). This is an interesting and novel topic. The methodology used is adequate and results presented in a clear and understandable manner. The Authors discuss their findings properly and provide previous bibliography. Overall, this is a well-written manuscript concerning an interesting topic 

Some minor corrections are required.

Please move the text in lines 112-113 to the Results section

Please provide CRP cutoff for exclusion of patients 

Reviewer 3 Report

I would like to congratulate the authors on the results of this small case-control trial. In general the article is well written, the results are promising and clear, but there are several issues that have to be addressed:

         1. In the abstract and was mentioned „typically developing“ as a desription for non-DS controls. I would prefer individuals without sign of DS or non-DS individuals to make it clear and not exclusive. Or healthy individuals as used in the latter text of the manuscript

          2. MPO, myeloperoxidase not myeloperoxidase

      3. The difference of the Lp (a) shold be adjusted for BMI since the difference in the DS and non-DS group is significant and may act as a confounder variable, especially in such a small cohort

Author Response

This manuscript is a resubmission of an earlier submission. The following is a list of the peer review reports and author responses from that submission.